

# Conditioned medium from M2b macrophages modulates the proliferation, migration, and apoptosis of pulmonary artery smooth muscle cells by deregulating the PI3K/Akt/FoxO3a pathway

Suiqing Huang[1,2,*], Yuan Yue[1,2,*], Kangni Feng[1], Xiaolin Huang[1,2], Huayang Li[1,2], Jian Hou[1,2], Song Yang[2,3], Shaojie Huang[1,2], Mengya Liang[1], Guangxian Chen[1] and Zhongkai Wu[1]

[1] Department of Cardiac Surgery, The First Affiliated Hospital of Sun Yat-sen University, Guangzhou, China
[2] NHC Key Laboratory of Assisted Circulation, Sun Yat-sen University, Guangzhou, China
[3] Department of Cardiosurgery Intensive Care Unit, The First Affiliated Hospital of Sun Yat-sen University, Guangzhou, China
[*] These authors contributed equally to this work.

Corresponding authors
Zhongkai Wu,
wuzhk@mail.sysu.edu.cn
Guangxian Chen, gx.chan@163.com

## ABSTRACT

**Background**. Immunity and inflammation are considered to be central features of pulmonary artery hypertension (PAH), in which macrophages are one of the main components of inflammatory cell infiltration around the pulmonary artery. M2b macrophages, which are different from M1 and M2 macrophages, are believed to have immunomodulatory activities and produce little fibrosis. The purpose of this study was to explore the effect of M2b macrophages on pulmonary artery smooth muscle cells (PASMCs) derived from monocrotaline-induced PAH rats.

**Methods**. PASMCs were cultured in serum-free medium, the supernatant of M0 macrophages, or the supernatant of M2b macrophages for 24 hours. Then cell proliferation was assessed by cell counting kit-8 and cell migration ability was detected by wound healing and transwell assays. The apoptosis rate of cells was determined by TUNEL staining and annexin V-PE/7-ADD staining. Western blot was used to detect the expression of Bcl-2 family proteins, cleaved caspase-9 and PI3K/Akt/FoxO3a pathway. LY294002 (a specific inhibitor of PI3K) was used to investigate its effect on PASMCs and its relationship with M2b macrophages.

**Results**. Conditioned medium from M2b macrophages significantly inhibited the proliferation and migration of PASMCs compared with the control group and M0 macrophage group. Furthermore, conditioned medium from M2b macrophages promote PASMC apoptosis and increased the expression of pro-apoptotic proteins Bax and cleaved caspase-9, inhibited the expression of anti-apoptotic proteins Bcl-2 and Bcl-xl. Finally, conditioned medium from M2b macrophages inhibited the PI3K/Akt/FoxO3a pathway. Inhibition of PI3K/Akt/FoxO3a pathway also significantly inhibit the proliferation, migration, and apoptosis resistance of PASMCs.

**Conclusion**. Conditioned medium from M2b macrophages can inhibit the proliferation, migration, and apoptosis resistance of PASMCs, which may be at least partially by deregulating the PI3K/Akt/FoxO3a pathway.

## INTRODUCTION

Pulmonary artery hypertension (PAH) is characterized by a progressive increase in pulmonary vascular resistance, identified by progressive hypoxia and right heart failure, which seriously affects the survival and prognosis of patients with cardiovascular and respiratory diseases (*Humbert, Sitbon & Simonneau, 2004*). Recently, with more in-depth research on the mechanism of PAH, it has been recognized that pulmonary vascular remodeling is the most crucial reason for the continuous development and deterioration seen in PAH and the failure of drug therapy (*Thompson & Lawrie, 2017*). Pulmonary vascular remodeling is mainly due to medial hypertrophy and neointimal lesion formation, and the apoptosis resistance, proliferation, and migration of pulmonary artery smooth muscle cells (PASMCs) play a vital role in this process. Therefore, exploring the molecular mechanism of PASMC proliferation, migration, and apoptosis resistance is an essential means to prevent pulmonary vascular remodeling, and thus, to treat PAH.

Since 1994, when *Tuder et al. (1994)* first found a large number of inflammatory cells infiltrating lung tissues in patients with PAH, an increasing number of studies have confirmed that inflammation and innate immunity play a vital role in the development of PAH (*Ogawa et al., 2011*; *Rabinovitch et al., 2014*; *Tang et al., 2015*; *Yang et al., 2018*). The synergistic action of various infiltrating inflammatory cells is also considered to be one of the critical factors involved in pulmonary vascular remodeling (*Rabinovitch et al., 2014*). Moreover, macrophages are one of the main components of inflammatory cell infiltration around the pulmonary artery during the development of PAH and are an essential source of local factors regulating pulmonary vascular remodeling. Studies have also shown that the phenotypic transformation and proliferation of PASMCs depend on macrophages (*Lee et al., 2012*; *Vergadi et al., 2011*), and some drugs targeting macrophages have achieved an excellent therapeutic effect in PAH. For example, *Frid et al. (2006)* have reported that selective depletion of monocyte/macrophage population by clodronate-liposomes or gadolinium chloride can prevent pulmonary adventitial remodeling and relieve pulmonary hypertension.

Macrophages are heterogeneous, and the surrounding microenvironment regulates their phenotype and function. Polarized macrophages can be roughly divided into three categories: classically activated macrophages (M1), alternatively activated macrophages (M2) and regulatory macrophages (M2b). M1 macrophages, which are typically induced by Toll-like receptor (TLR) ligands (bacterial lipopolysaccharide) or by some cytokines, exert cellular immunity by releasing a large number of inflammatory factors to eliminate
pathogens. Nevertheless, they mediate reactive oxygen species (ROS)-induced tissue damage, and excessive inflammatory reactions can also damage healthy tissues (*Arora et al., 2018*; *Bashir et al., 2016*). M2 macrophages, also known as alternatively activated macrophages, can be polarized by type 2 T-helper cell (Th2) cytokines (IL-4, IL-13, and IL-33), which can produce a variety of anti-inflammatory factors and high levels of TGF-β. M2 macrophages inhibit inflammation and promote angiogenesis but encourage the deposition of extracellular matrices and the formation of collagen fibers, leading to tissue fibrosis (*Murray & Wynn, 2011*; *Shapouri-Moghaddam et al., 2018*; *Wynn & Vannella, 2016*). M2b macrophages are defined as regulatory macrophages. They can be induced upon combined exposure to immune complex and TLR agonists or by IL-1R agonists. Moreover, their markers include IL-10, CCL1, LIGHT, CD86, SPHK1, TNF-α, and IL-6 (*Wang et al., 2019*). M2b macrophages not only secrete pro-inflammatory factors (IL-1β, IL-6, and TNF-α) but also express and secrete large amounts of the anti-inflammatory cytokine IL-10 and low levels of IL-12 (*Shapouri-Moghaddam et al., 2018*; *Wang et al., 2019*). They balance anti-inflammatory and pro-inflammatory functions and do not cause deposition of the extracellular matrix.

Since M2b macrophages have immunomodulatory activities and produce little fibrosis, we hypothesize that M2b macrophages can play a therapeutic role in PAH. In this study, we isolated PASMCs from monocrotaline (MCT)-induced PAH rats. We then induced the polarization of M2b macrophages in vitro and incubated the supernatant of M2b macrophages, M0 macrophages, and serum-free culture medium with PASMCs to evaluate their effects on cell proliferation, migration, and apoptosis. To explore the possible mechanism, the effects of M2b macrophages on the PI3K/Akt/FoxO3a pathway were examined. The effects of inhibiting the PI3K/Akt/FoxO3a pathway on the proliferation, migration, and apoptosis of pulmonary artery smooth muscle cells were also investigated.

## MATERIALS & METHODS

### Animals

Sprague-Dawley (SD) rats (male, 6- to 8-week-old, weight 200–250 g) were obtained from the Laboratory Animal Center of Guangzhou University of Chinese Medicine (Guangzhou, China, SCXK-2013-0020). At present, rats are the main animals used to establish animal models of pulmonary artery hypertension. Animals were maintained in cages under constant temperature (22 ± 2 °C), humidity (45 ± 5%), and a 12 h day and 12 h night cycle. They were given standard rodent chow and water ad libitum. Twelve rats were used for the isolation of pulmonary artery smooth muscle cells, and another 16 were used for the isolation of bone marrow macrophages (BMDMs). All animal procedures were approved by the Animal Ethical and Welfare Committee of Sun Yat-sen University (IACUC approval number: DB-17-0506).

### Pulmonary artery smooth muscle cells isolated from MCT-induced PAH rats

Rat PASMCs were isolated from the pulmonary trunk of adult male SD PAH rats 28 days after a single subcutaneous injection of 60 mg/kg MCT (Sigma Aldrich, St. Louis, MO,

USA) at the age of 6–8 weeks. The rats were anesthetized with pentobarbital injection (120 mg/kg, i.p.). Then the thorax was opened and the pulmonary artery was separated from cardiopulmonary tissue under aseptic conditions. The outer membrane and endothelium of the pulmonary artery were carefully scraped off. The remaining smooth muscle was cut using ophthalmic scissors into 1 mm$^3$ tissue fragments and placed in a 25 ml culture flask in Dulbecco's modified Eagle's medium/high glucose (DMEM, Gibco, Grand Island, NY, USA) containing 15% fetal bovine serum (Gibco), 1% penicillin-streptomycin (Gibco) for cultivation (complete DMEM). The tissue fragments were placed in an incubator (5% $CO_2$, 37 °C), and the complete DMEM was changed three days later. Cells grew from the tissue sample after five days. The cells were fed with complete DMEM every two days; when they reached 80% confluence, they were subcultured with 0.25% trypsin (Gibco). The identification and purity verification of PASMCs was accomplished by immunofluorescence staining of α-smooth muscle actin (α-SMA) and smooth muscle myosin heavy chain (SM-MHC). The PASMCs were studied at the early-passage stage (passages 2–5).

## Immunofluorescence staining assay

Immunofluorescence staining against α-SMA and SM-MHC was used to determine the purity of the PASMCs. The cells were fixed with 4% paraformaldehyde and permeabilized in 0.2% Triton X-100. After blocking with 1% BSA, the cells were incubated with primary antibodies, including rabbit monoclonal antibody against α-SMA (Abcam, Cambridge, MA, USA) and mouse monoclonal antibody against SM-MHC (Abcam) at 4 °C overnight. The cells were then incubated with Alexa Fluor 488-conjugated donkey anti-rabbit secondary IgG (Invitrogen, Carlsbad, CA, USA) and Alexa Fluor 568-conjugated donkey anti-mouse secondary IgG (Invitrogen). The nuclei were stained with DAPI (Sigma Aldrich). Fluorescence was observed with an immunofluorescence microscope (Carl Zeiss, Jena, Germany).

## Isolation and *in vitro* polarization of macrophages

M2b macrophages were differentiated from BMDMs of adult male SD rats. The rats used for macrophage extraction were sacrificed by cervical dislocation. DMEM was used to wash the bone marrow cavity of the femur and tibia to collect the bone marrow. After centrifugation (500 g for 5 min), the cells were cultured in flasks in DMEM containing 10% fetal bovine serum, 1% penicillin-streptomycin and 10 ng/ml macrophage colony-stimulating factor (MCSF, PeproTech, Rocky Hill, NJ, USA). On the second and fourth day after establishing the initial culture, the cells were confirmed to adhere by an inverted phase-contrast bright-field microscope and showed slight branching. Medium containing non-adherent cells was discarded. The cells were washed once with DMEM, and MCSF culture medium was added. After six days of culturing, the cells grew into mature BMDMs. The BMDMs were replated and differentiated into M2b macrophages after the addition of 50 μg/ml IgG (Sigma Aldrich) and 100 ng/ml LPS (Sigma Aldrich) (*Graff et al., 2012*). M0 macrophages do not require the addition of stimulating factors. After 24 h of stimulation, the culture medium of the cells was removed, and fresh medium without stimulation agents was

applied to further culture the cells for 24 h to collect the secretory substances of the M0 and M2b macrophages. Cell-free supernatants were then collected at 24 h for coincubation with PASMCs *in vitro*. The stimuli for PASMCs were derived from the supernatant secreted by the equal number of macrophages to PASMCs in each experiment (CCK-8 assay, wound healing assay, transwell assay, TUNEL assay, annexin V-PE/7-ADD staining assay, and Western blot).

## Identification of M2b macrophages by flow cytometry

Cells were prepared as single-cell suspensions. They were conjugated with rabbit anti-rat LIGHT primary antibody (Abcam) or isotype control (Abcam) and then with Alexa Fluor 488-conjugated donkey anti-rabbit IgG (Invitrogen). Afterwards the cells were combined with APC A750-rat CD45 (eBioscience, San Diego, CA, USA) or isotype control (eBioscience). Finally, flow cytometry was performed with a Beckman Coulter CytoFLEX flow cytometer (Beckman Coulter, Miami, FL, USA), and the results were analyzed with FlowJo software.

## Quantitative real-time PCR (qRT-PCR)

Total RNA was extracted from tissue homogenates using TRIzol reagent (Invitrogen). Next, 200 µl of chloroform was used for phase separation, and 100% isopropanol was applied to precipitate the RNA. Finally, after washing twice with 75% ethanol, the RNA was eluted in 30 µl of RNase-free water. The concentration of RNA was measured using a NanoDrop 2000 spectrophotometer (Thermo Fisher Scientific, Waltham, MA, USA). Reverse transcription of 1000 ng of total RNA per reaction was performed with the PrimeScript RT Master Mix (Takara, Tokyo, Japan). qRT-PCR was then performed on a Light Cycler 480 system (Roche, Basel, Switzerland) using TB Green Premix Ex Taq II (Takara), according to the manufacturer's instructions. The comparative threshold cycle (CT) value for housekeeping gene GAPDH was used to normalize the loading variations in the PCR. The list of primers with their sequences is as followed: induced CCL-1 Fw: AGAGCCTGCAGTTTCACTCA, Rev: GATCTGTGAGCCTGCATCAGT; IL-10: Fw: GGAGCAGGTGAAGAATGAT, Rev: TCTCGTAGGCTTCTATGCAGTTG; and GAPDH Fw: GGTCATCCATGACAACTT, Rev: GGGGCCATCCACAGTCTT.

## CCK-8 assay

Cell proliferation was assessed with a CCK-8 assay (Beyotime Biotechnology, Shanghai, China), according to the manufacturer's instructions. PASMCs were serum starved in serum-free medium for 24 h prior to the proliferation experiments. Approximately $1 \times 10^5$ PASMCs were plated in 96-well plates. After different stimuli were added to the cells, they were incubated in an incubator (5% $CO_2$, 37 °C) for 24 h. Fresh serum-free DMEM solution containing 10 µl of CCK-8 solution was then added to each well and incubated at 37 °C for 1 h in the dark. Analyses were performed with a modular multitechnology microplate reader (Thermo Fisher Scientific) by measuring the optical density value at 450 nm.

## Wound healing assay

A wound healing assay was used to assess the migration of PASMCs. Approximately $4 \times 10^5$ cells were seeded in a six-well plate and cultured in complete DMEM medium to confluence. The tip of a 200 μl pipette was directly scratched through the cell monolayer to obtain a wide noncellular area, and the well was washed three times with PBS to remove cell debris caused by the scratch. The cover of the 6-well plate was marked to ensure that the field of view was the same in the photos. The medium was then replaced with medium containing different stimuli. The scratches of each group were observed under an inverted phase-contrast microscope at 0 h, 12 h, and 24 h. The area of scratch was measured using ImageJ software (National Institutes of Health, USA).

## Transswell assay

The transwell assay was performed as previously described (*Qiu et al., 2020*). Briefly, 500 μl of culture medium containing different stimuli was placed in a 24-well tissue culture plate, and a transwell (8 μm pore size, Corning, NY, USA) insert was transferred to the plate. After preparing the cell suspensions ($1 \times 10^5$ cells/ml), $5 \times 10^4$ PASMCs were added to the top chamber in serum-free culture medium. After 24 h of incubation, to quantify migration through the porous membrane, PASMCs on the top side of the membrane were wiped off with a cotton swab. The cells invading through the membrane were then fixed with 4% paraformaldehyde for 30 min and stained with 0.1% crystal violet for 15 min. The cells were observed with a microscope camera. For each filter, five randomly selected fields were imaged to assess the migrated cells.

## TUNEL assay

DNA fragmentation analysis was carried out by using a TUNEL fluorescein assay kit (Roche, Basel, Switzerland). PASMCs were seeded onto 24-well plates with a circle microscope cover glass (15 mm) at a concentration of $3 \times 10^5$ cells per well. After treatment, PASMCs were washed three times with PBS, fixed with 4% paraformaldehyde for 1 h, and permeabilized with 0.1% Triton X-100 for 5 min. The cells were then incubated with TUNEL reaction mixture (terminal deoxynucleotidyl transferase and nucleotide mixture) for 1 h at 37 °C in a humidified and dark atmosphere. After a rinse with PBS, the samples were analyzed under a Carl Zeiss LSM 800 Meta confocal laser scanning microscope (Carl Zeiss).

## Annexin V-PE/7-ADD staining assay

The apoptotic ratio of PASMCs cells was also determined using an annexin V-PE/7-ADD apoptosis detection kit (BD Bioscience, San Diego, CA, USA). After different treatments, $1 \times 10^5$ cells were harvested, washed with PBS, and stained with annexin V-PE and 7-ADD according to the manufacturer's instructions. Up to $1 \times 10^4$ cells per sample were recorded using a Beckman CytoFLEX flow cytometer and analyzed with FlowJo software. Annexin V-PE-positive and 7-ADD-negative cells were considered to be apoptotic cells.

## Western blot analysis

After 24 h of stimulation, $4 \times 10^5$ PASMCs were collected and lysed with RIPA lysis buffer (Beyotime Biotechnology) that contained a protease and phosphorylase inhibitor

cocktail (Merck Millipore, Billerica, MA, USA). The protein concentration was determined using a Bio-Rad protein assay kit (Bio-Rad Laboratories, CA, USA). The whole lysate was fractionated by 10% to 12% SDS-PAGE and transferred to PVDF membranes (Merck Millipore). The membranes were blocked and then incubated overnight at 4 °C with primary antibodies against Bax (Cell Signaling Technology, Danvers, MA, USA), Bcl-2 (Abcam), Bcl-xl (Cell Signaling Technology), cleaved caspase-9 (Cell Signaling Technology), phosphor-PI3K (Abcam), PI3K (Cell Signaling Technology), phospho-Akt (Cell Signaling Technology), Akt (Cell Signaling Technology), phosphor-FoxO3a (Abcam), FoxO3a (Abcam) and GAPDH (Proteintech, Rosemont, IL, USA). The membranes were then incubated with the respective peroxidase-conjugated secondary antibodies (SouthernBiotech, Birmingham, AL, USA) for 1 h. Finally, the proteins were visualized using Western chemiluminescent HRP substrate (Merck Millipore) and ChemiDoc Touch (Bio-Rad Laboratories).

### LY294002 for phosphoinositide 3-kinases (PI3K) inhibition

LY294002 (in DMSO vehicle, Sigma Aldrich) was applied to inhibit PI3K activity. The control group had the same volume of DMSO added to it. At 24 h after seeding, the PASMCs were treated for 24 h at 37 °C in 5% $CO_2$ with different cell supernatants containing 30 $\mu$M LY294002 to inhibit PI3K activity. At 24 h after seeding, the PASMCs were treated for 24 h at 37 °C in 5% CO2 with different cell supernatants containing 30 $\mu$M LY294002 to inhibit PI3K activity, and then subjected to Western blot analysis, CCK-8 assay, transwell assay, and annexin-V/PE staining assay.

### Statistical analysis

Values are presented as the mean $\pm$ SEM. Statistical analysis was performed using SPSS 22.0 software. The differences between two groups were compared by two-tailed Student's $t$-test. Comparisons among more than two groups were made with one-way ANOVA with Tukey post hoc test. A value of $p < 0.05$ was considered to represent a statistically significant difference between groups.

## RESULTS

### Identification of PASMCs from MCT-induced PAH rats and M2b macrophages

PASMCs were isolated from MCT-induced PAH rats. Immunofluorescence staining showed that 94.52 $\pm$ 1.76% of cells co-expressed $\alpha$-SMA and SM-MHC (Figs. 1A–1D). Flow cytometry was used to assess the purity of the M2b macrophages. The results showed that >90% of the cells were LIGHT+ CD45+ after stimulation, indicating most of bone marrow cells were polarized into M2b macrophages (Figs. 1E and 1F). qRT-PCR was also used to identify the phenotypes of the polarized macrophages. The expression of CCL-1 and IL-10 were significantly higher in the M2b macrophages compared to M0 macrophages (both $p < 0.05$, Figs. 1G and 1H).

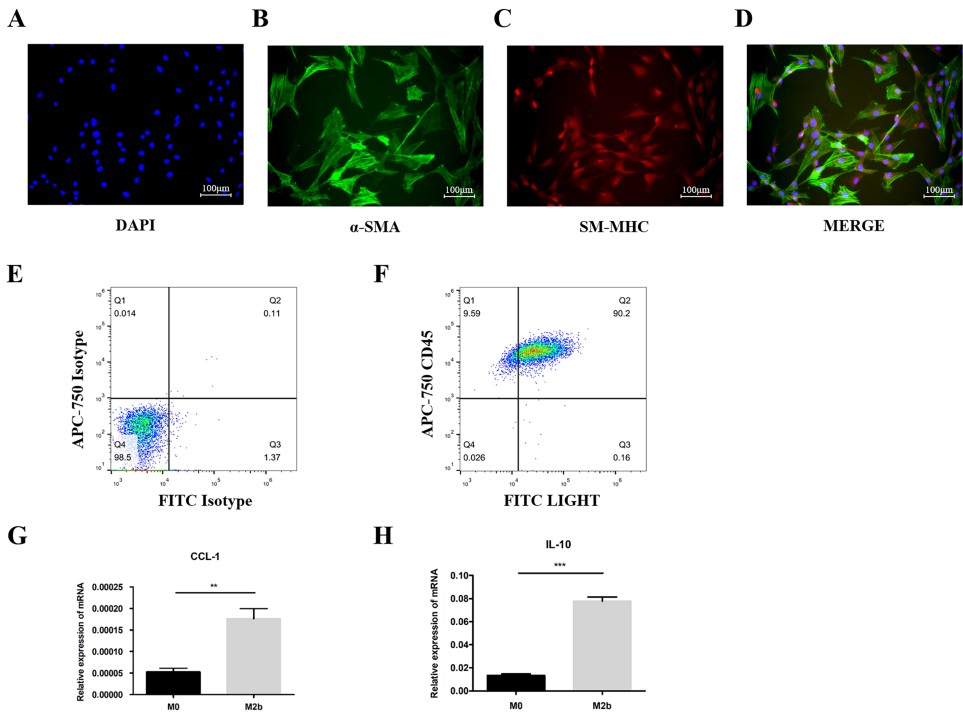

**Figure 1 Identification of PASMCs from monocrotaline-induced pulmonary artery hypertension rats and M2b macrophages.** (A–D) Immunofluorescence experiments for co-staining of α-SMA and SM-MHC. Quantitative results showed that $94.52 \pm 1.76\%$ of cells co-expressed α-SMA and SM-MHC ($n =$ 3). (E, F) M2b macrophages were stained to assess LIGHT and CD45 expression and were analyzed by flow cytometry. Over 90% of the cells were LIGHT+ CD45+ ($n = 3$). (G, H) The mRNA levels of CCL-1 and IL-10 were detected by qRT-PCR in macrophages. Data are shown as the mean $\pm$ SEM ($n = 3$ for each group). $**p < 0.01$, $***p < 0.001$. "M0" indicates the M0 macrophage group, "M2b" indicates the M2b macrophage group.

## Conditioned medium from M2b macrophages inhibited PASMC proliferation and migration

To evaluate the role of M2b macrophages in the proliferation of PASMCs derived from MCT-induced PAH rats, we used the cell counting kit-8 (CCK-8) assay to assess proliferation (Fig. 2A). PASMCs were cultured in serum-free medium (control group), the supernatant of M0 macrophages (M0 group), or the supernatant of M2b macrophages (M2b group) for 24 h before CCK-8 detection. The proliferation of PASMCs in the M2b group was significantly lower than those in the control group ($p < 0.01$) and the M0 group ($p < 0.001$). To elucidate the anti-migratory effects of different macrophages, we performed two types of migration assays. By using the wound-healing assay, it was observed that conditioned medium from M2b macrophages significantly inhibited the migration of PASMCs compared with the control and M0 groups (both $p < 0.001$, Figs. 2B–2K). In the transwell assay (Figs. 2L–2O), the number of migrated cells decreased significantly at 24 h in the M2b group compared with the control group ($p < 0.01$) and M0 group ($p < 0.001$). Thus, these observations suggest that conditioned medium from M2b macrophages can inhibit the proliferation and migration of PASMCs.

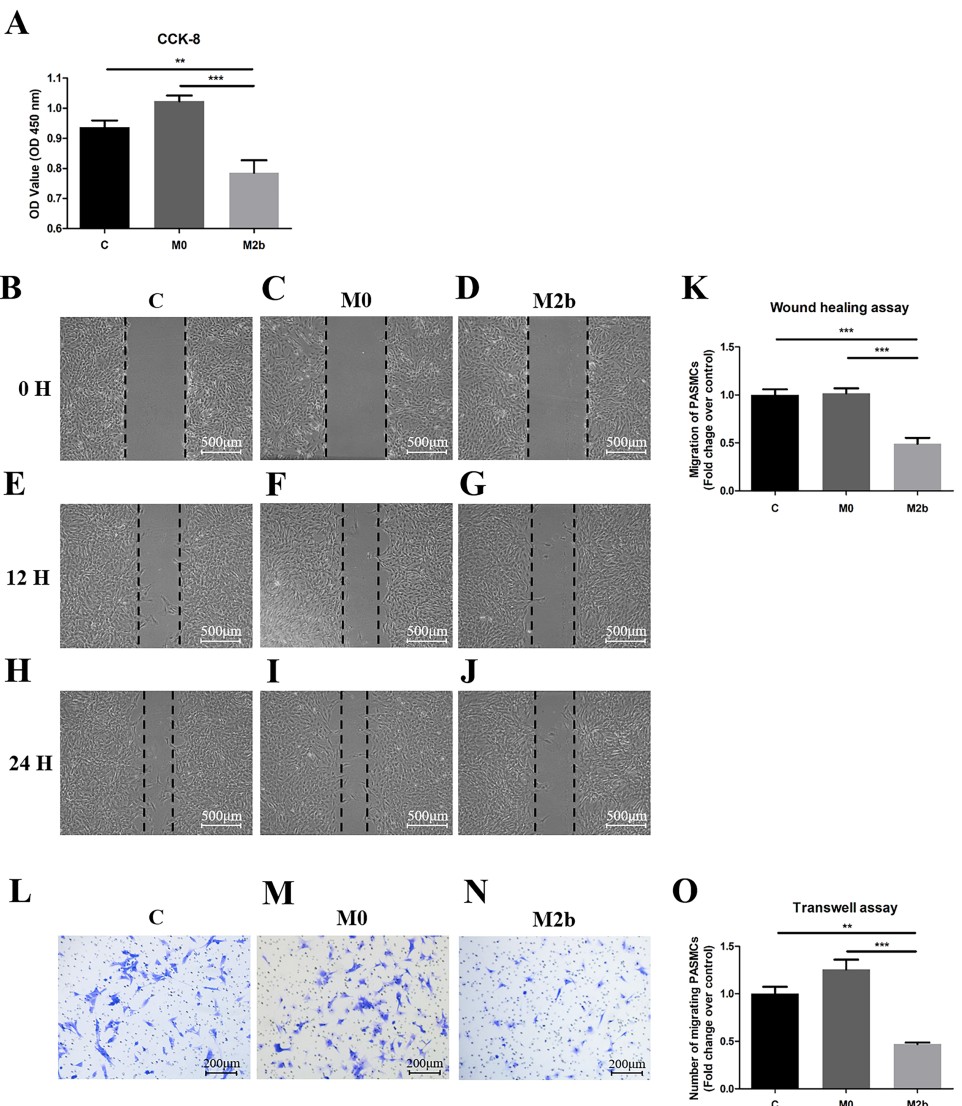

**Figure 2** **Conditioned medium from M2b macrophages inhibited the proliferation and migration of PASMCs.** PASMCs were treated with serum-free medium, supernatant of M0 macrophages, and supernatant of M2b macrophages for 24 hours, and the proliferation and migration of the different groups were detected. (A) Proliferation of PASMCs detected by CCK-8 assay ($n = 6$ for each group). (B–K) Migration of PASMCs detected by wound healing assay ($n = 5$ for each group, original magnification: 50×). (L–O) Migration of PASMCs detected by transwell assay ($n = 5$ for each group, original magnification: 100×). Data are shown as the mean ± SEM. **$p < 0.01$, ***$p < 0.001$. "C" indicates the control group, "M0" indicates the M0 macrophage group, "M2b" indicates the M2b macrophage group.

## The effect of conditioned medium from M2b macrophages on PASMC apoptosis

We further investigated the effect of conditioned medium from M2b macrophages on apoptosis of PASMCs. Apoptotic nuclei were detected by TdT-mediated dUTP nick end labeling (TUNEL) staining with *in situ* cell death detection kit, and all nuclei were identified by DAPI staining. The DAPI staining showed that most of the nuclei in the control and M0

groups were complete and oval in shape. However, some of the nuclei in the M2b group were rippled or creased and some nuclei were cleaved into fragments, producing apoptotic bodies and presenting typical apoptosis. TUNEL-positive cells were visualized as indicated by red fluorescence staining, and the percentage of apoptotic cells was determined by the ratio of the number of TUNEL-positive PASMCs to the total number of cells. As shown in Figs. 3A–3J, the apoptosis rate of the M2b group was significantly higher than those of the control and M0 groups (both $p < 0.001$). Flow cytometric analysis with annexin V-PE and 7-ADD staining was used to determine the effect of M2b macrophages on PASMC apoptosis. Early apoptotic cells were annexin V (+) and 7-ADD (-). As shown in Figs. 3K and 3L, the apoptosis rate of PASMCs treated with serum-free medium was $6.24 \pm 2.06\%$, while that of the M0 group was $6.39 \pm 1.82\%$, showing no significant difference between the two groups ($p > 0.05$). However, the M2b group showed a markedly increased apoptotic cell percentage ($14.97 \pm 1.12\%$) compared to the other two groups (both $p < 0.01$).

### Bcl-2 family proteins and cleaved caspase-9 function during apoptosis

Bcl-2 family proteins and cleaved caspase-9 regulate mitochondrial-dependent apoptotic signaling. To further validate the molecular mechanism of apoptosis, the expression of Bcl-2 family proteins and cleaved caspase-9 in M2b macrophage conditioned medium-treated PASMCs was determined by Western blot analysis. The pro-apoptotic protein expression of Bax and cleaved caspase-9 was significantly increased, the anti-apoptotic proteins Bcl-2 and Bcl-xl were significantly decreased, and the ratio of Bax/Bcl-2 was significantly increased in the M2b group compared with the other two groups (all $p < 0.05$, Figs. 3M–3R). These results suggest that Bcl-2 family proteins and cleaved caspase-9 may be involved in the conditioned medium from M2b macrophage induction of PASMC apoptosis.

### Conditioned medium from M2b macrophages inhibited the PI3K/Akt/FoxO3a signaling pathway

To investigate the role of the PI3K/Akt/FoxO3a pathway in mediating the effects of conditioned medium from M2b macrophages in PASMCs, we used Western blot to detect protein phosphorylation of PI3K, Akt, and FoxO3a. As shown in Figs. 4A–4D, PASMCs maintained an activated state of PI3K/Akt/FoxO3a in the control and M0 groups. However, a significant decrease in the levels of the p-PI3K (Y607), p-Akt (Ser473), and p-FoxO3a (Ser253) was observed in the M2b group compared with the other two groups (all $p < 0.05$). These results suggest that conditioned medium from M2b macrophages induces the downregulation of the PI3K/Akt/FoxO3a signaling pathway in PASMCs.

### Conditioned medium from M2b macrophages may inhibit the proliferation and migration of PASMCs by deregulating the PI3K/Akt/FoxO3a pathway

To examine the effects of conditioned medium from M2b macrophages and the PI3K/Akt/FoxO3a pathway on the cell proliferation and migration of PASMCs, we used LY294002 (a specific inhibitor of PI3K) to investigate its effect on PASMCs and its relationship with M2b macrophages. As shown in Figs. 5A–5D, both LY294002 (30 μM)

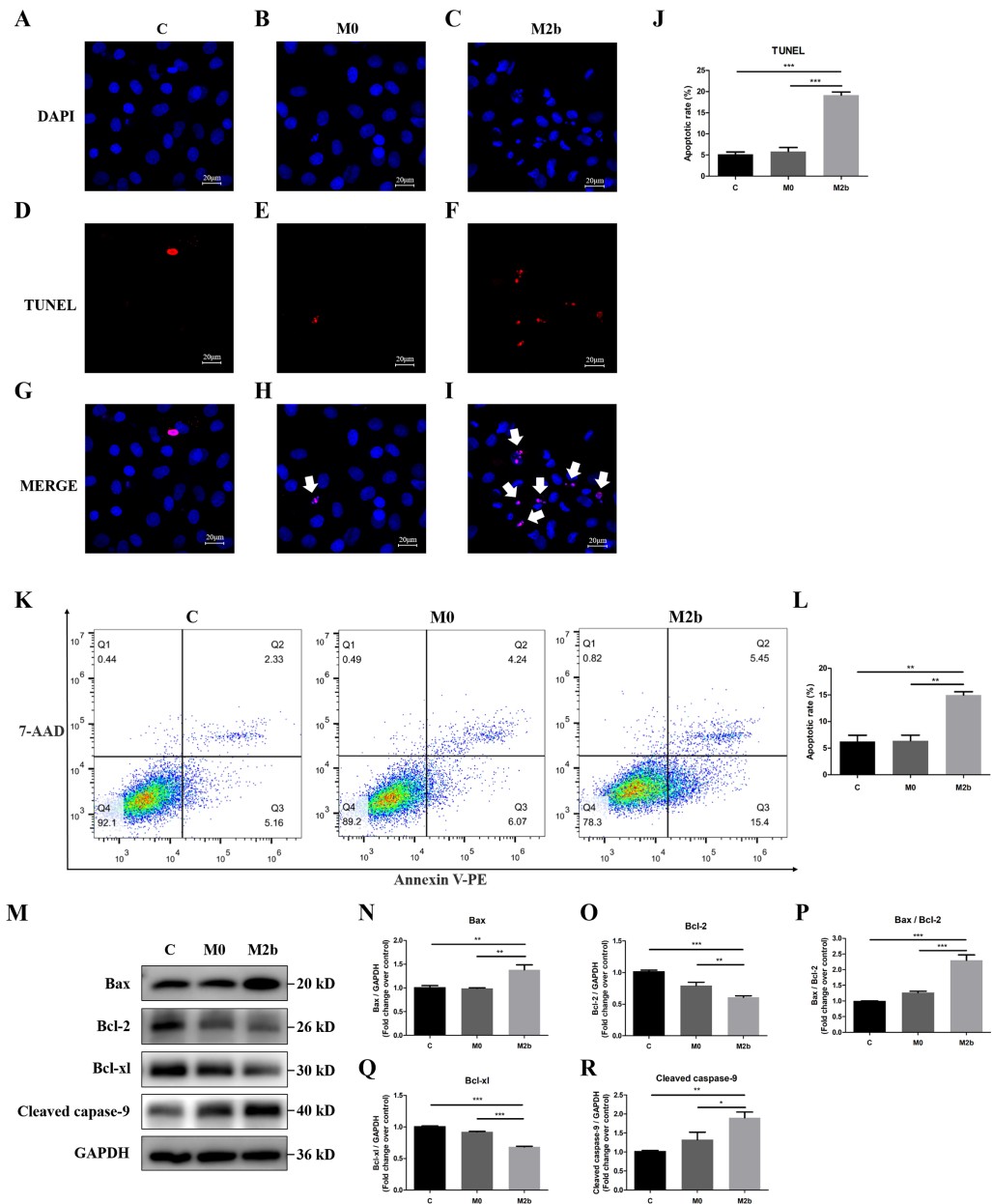

**Figure 3 Conditioned medium from M2b macrophages attenuated the apoptosis resistance of PASMCs.** PASMCs were treated with serum-free medium, the supernatant of M0 macrophages or the supernatant of M2b macrophages for 24 hours before detection. (A–J) TUNEL staining was used to assess the apoptosis of PASMCs ($n = 5$ for each group, original magnification: $200\times$). The total number of PASMCs counted across the $n = 5$ was as follows: C group = 207, M0 group = 177, M2b group = 188. The white arrow points to rippled or creased nuclei. The apoptotic cell proportion was calculated as the ratio of TUNEL-positive cells to the total number of PASMCs. (K, L) Annexin V-PE/7-ADD staining was used to assess the apoptosis of PASMCs ($n = 3$ for each group). The number of apoptotic cells was quantified by flow cytometry after the cells were stained with annexin V-PE and 7-ADD. Early apoptotic cells were determined by counting the percentage of annexin V (+), 7-ADD (−). (M–R) Western blot was then used to assess the protein expression of Bax, Bcl-2, Bcl-xl, cleaved caspase-9, and GAPDH ($n = 4$ for each group). Data are shown as the mean ± SEM. $*p < 0.05$, $**p < 0.01$, $***p < 0.001$. "C" indicates the control group, "M0" indicates the M0 macrophage group, "M2b" indicates the M2b macrophage group.

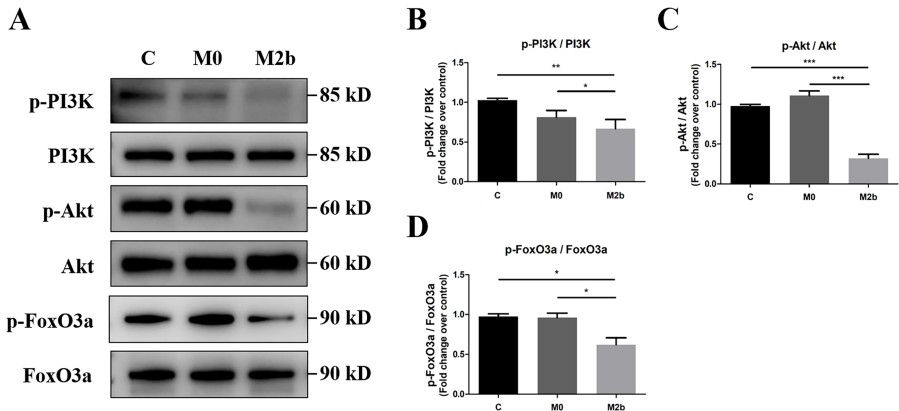

**Figure 4 Conditioned medium from M2b macrophages inhibited the PI3K/Akt/FoxO3a signaling pathway.** PASMCs were treated with serum-free medium, the supernatant of M0 macrophages or the supernatant of M2b macrophages for 24 hours. (A–D) The expression and phosphorylation of PI3K, Akt, and FoxO3a was then assessed by Western blot ($n = 4$ for each group). Data are shown as the mean $\pm$ SEM. *$p < 0.05$, **$p < 0.01$, ***$p < 0.001$. "C" indicates the control group, "M0" indicates the M0 macrophage group, "M2b" indicates the M2b macrophage group.

and conditioned medium from M2b macrophages significantly reduced the protein levels of p-PI3K, p-Akt, and p-FoxO3a, and the protein expression levels were even lower when both of them acted on the cell simultaneously (all $p < 0.05$). The cell viability was determined by measuring CCK-8. The PI3K inhibitor suppressed cell proliferation compared to the control group ($p < 0.001$), which was also found for PASMCs treated with the supernatant of M2b macrophages. Cell viability was further suppressed in PASMCs treated with both the supernatant of M2b macrophages and LY294002 compared to PASMCs treated with the supernatant of M2b macrophages or LY294002 alone (both $p < 0.05$, Fig. 5E). The transwell assay results showed that LY294002 inhibited the migration of PASMCs as much as the M2b macrophage supernatant did. The number of migrated cells was even lower under the joint action of LY294002 and M2b macrophages compared with the PASMCs treated with the supernatant of M2b macrophages or LY294002 alone (both $p < 0.05$, Figs. 5F–5J). In summary, these results suggest that conditioned medium from M2b macrophages may inhibit the proliferation and migration of PASMCs by deregulating the PI3K/Akt/FoxO3a pathway.

## Conditioned medium from M2b macrophages may inhibit apoptosis resistance in PASMCs by deregulating the PI3K/Akt/FoxO3a signaling pathway

To further investigate whether the PI3K/Akt/FoxO3a pathway participates in M2b macrophage-mediated apoptosis, LY294002 (30 μM) was used to inhibit PI3K. Apoptosis was detected by annexin V-PE/7-ADD staining, and the percentage of apoptotic cells was $11.77 \pm 0.82\%$ in PASMCs treated with LY294002 and $14.50 \pm 0.26\%$ in cells treated with the supernatant of M2b macrophages alone, which was significantly increased compared with cells in the control group ($7.19 \pm 0.25\%$, both $p < 0.05$). There were no significant

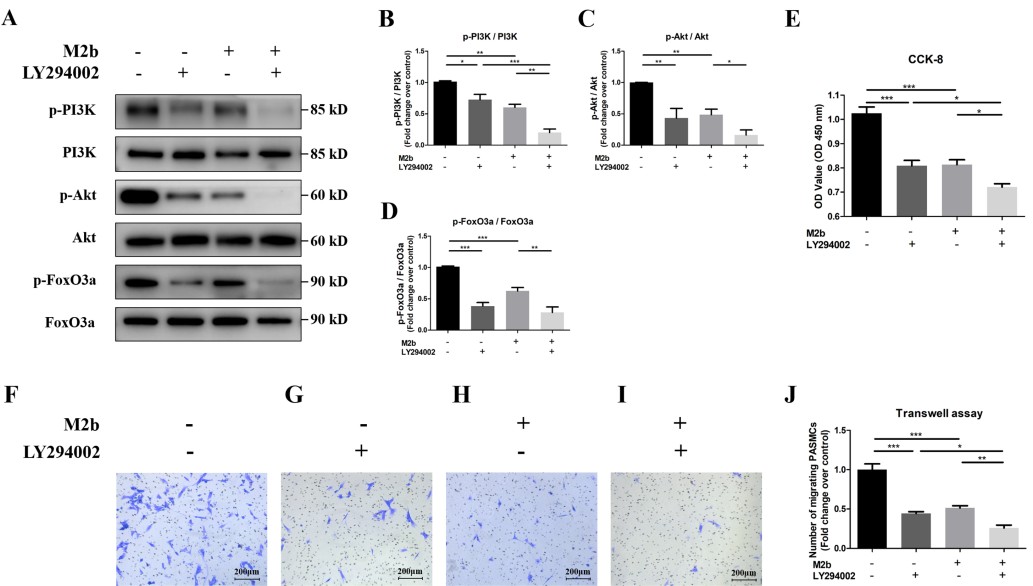

**Figure 5** **Conditioned medium from M2b macrophages may inhibit the proliferation and migration of PASMCs by deregulating the PI3K/Akt/FoxO3a signaling pathway.** PASMCs were treated with serum-free medium, LY294002 (30 μM), the supernatant of M2b macrophages, or a combination of LY294002 (30 μM) and the supernatant of M2b macrophages for 24 hours. The PI3K/Akt/FoxO3a pathway activation, proliferation, and migration of different groups were detected. (A–D) Expression and phosphorylation of PI3K, Akt, FoxO3a assessed by Western blot analysis ($n = 4$ for each group). (E) Proliferation of PASMCs was detected by CCK-8 assay ($n = 5$ for each group). (F–J) Migration of PASMCs was detected by transwell assay ($n = 5$ for each group, original magnification: 100×). Data are shown as the mean ± SEM. *$p < 0.05$, **$p < 0.01$, ***$p < 0.001$.

differences in the apoptosis rates between the group treated with LY294002 and the group treated with the supernatant of M2b macrophages alone. Furthermore, compared with these two groups, PASMCs treated with both the M2b supernatant and LY294002 showed a markedly increased apoptotic cell percentage, at 18.93 ± 1.57% (both $p < 0.05$, Figs. 6A and 6B). To further confirm that PI3K/Akt/FoxO3a indeed regulates the apoptotic pathway through the regulation of Bcl-2 family proteins and cleaved caspase-9, we also used a PI3K inhibitor. We examined the protein expression of the Bcl-2 family and cleaved caspase-9 proteins with Western blot. The results showed that LY294002 and conditioned medium from M2b macrophages both reduced the protein expression levels of Bcl-2 and Bcl-xl and increased the expression of Bax and cleaved caspase-9 (all $p < 0.05$). The ratio of Bax/Bcl-2 was significantly increased ($p < 0.05$). Compared to the treatment of LY294002 alone, PASMCs treated with both LY294002 and M2b supernatant reduced the protein expression of Bcl-2 and Bcl-xl, and increased the expression of cleaved caspase-9 and the ratio of Bax/Bcl-2 (all $p < 0.05$). Compared to treatment with M2b supernatant alone, PASMCs treated with both LY294002 and M2b supernatant showed reduced protein expression of Bcl-xl and an increased ratio of Bax/Bcl-2 (both $p < 0.05$). Other expression differences were not statistically significant (Figs. 6C–6H). These results suggest that the inhibition of PI3K/Akt/FoxO3a may be an initial factor for Bcl-2 family-related apoptosis.

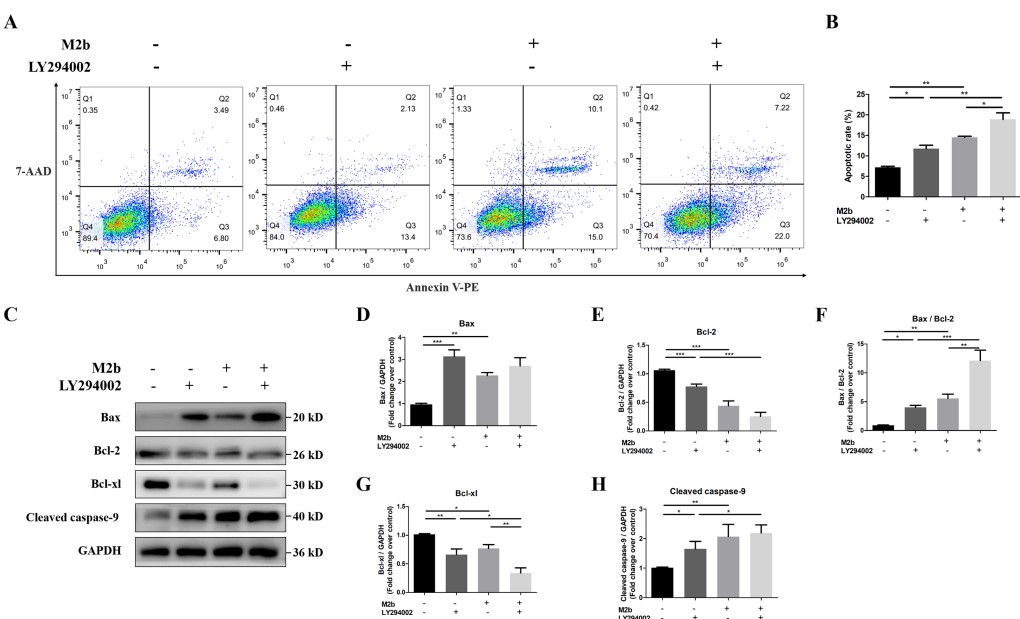

**Figure 6 Conditioned medium from M2b macrophages may inhibit apoptosis resistance in PASMCs by deregulating the PI3K/Akt/FoxO3a pathway.** PASMCs were treated with serum-free medium, LY294002 (30 μM), the supernatant of M2b macrophages, or a combination of LY294002 (30 μM) and the supernatant of M2b macrophages for 24 hours. The apoptosis and expression levels of Bcl-2 family proteins and cleaved caspase-9 in different groups were detected. (A, B) Annexin V-PE/7-ADD staining was used to assess the apoptosis of PASMCs ($n = 3$ for each group). (C–H) Western blot was used to assess the protein expression of Bax, Bcl-2, Bcl-xl, cleaved caspase-9, and GAPDH ($n = 4$ for each group). Data are shown as the mean ± SEM. *$p < 0.05$, **$p < 0.01$, ***$p < 0.001$.

In summary, these data suggest that conditioned medium from M2b macrophages may inhibit the apoptosis resistance of PASMCs via inhibiting the PI3K/Akt/FoxO3a pathway. In this process, the PI3K/Akt/FoxO3a pathway may regulate apoptosis through the Bcl-2 family proteins.

## DISCUSSION

In the present study, we assessed whether conditioned medium from M2b macrophages could have protective effects against the development of pulmonary artery hypertension. Our results showed that conditioned medium from M2b macrophages significantly inhibited the proliferation and migration of PASMCs from MCT-induced PAH rats. Moreover, after treatment with the supernatant of M2b macrophages, the apoptosis rate of PASMCs was increased, and Bcl-2 family proteins were involved in this process. Furthermore, conditioned medium from M2b macrophages deregulated the PI3K/Akt/FoxO3a signaling pathway, indicating that they might inhibit the proliferation, migration, and apoptosis resistance of PASMCs through this pathway.

Pulmonary artery hypertension still cannot be cured, prevented, or reversed, and a growing number of studies suggest that this is because current treatments do not improve pulmonary vascular remodeling (*Perrin et al., 2015*). The abnormal proliferation and

apoptosis resistance of PASMCs in the vascular wall of the distal pulmonary artery is the hallmark of pulmonary vascular remodeling in PAH (*Tuder, 2009*). Therefore, an effective treatment of PAH may depend on the development of new strategies to inhibit PASMC proliferation and apoptosis resistance.

Immunity and inflammation are increasingly recognized as central features of PAH. The synergistic effect of various infiltrating inflammatory cells, including T lymphocytes, B lymphocytes, dendritic cells, mast cells, monocytes, and macrophages, plays a vital role in pathogenesis (*Kuebler, Bonnet & Tabuchi, 2018*; *Willis et al., 2018*). It has been proved that monocytes and macrophages are the central effector cells that cause local pulmonary inflammation in PAH (*Florentin & Dutta, 2017*; *Frid et al., 2005*). Thus, many drugs that target monocytes and monocyte progenitors in the bone marrow can prevent pulmonary vascular remodeling (*Frid et al., 2006*; *Jin et al., 2006*). Furthermore, the reduction or depletion of macrophages can relieve PAH (*Tian et al., 2013*; *Zaloudikova et al., 2016*). Recent studies have highlighted the impact of microenvironments in PAH on the differentiation of recruited monocytes. In animal models of hypoxia-induced pulmonary artery hypertension, M1 macrophages are believed to be involved in the early process of injury occurrence, as well as in immune response and regulation (*Pugliese et al., 2017*). M2 macrophages play a role in promoting fibrosis and angiogenesis in pulmonary diseases (*Mora et al., 2006*). These alternatively activated macrophages promote the proliferation of PASMCs *in vitro* and are associated with the development of PAH *in vivo*. Therefore, they may play an essential role in the later development of pulmonary vascular remodeling (*Vergadi et al., 2011*). However, M2b macrophages are significantly different from M1 and M2 macrophages. M2b macrophages are regulatory macrophages that maintain a balance between anti-inflammatory and pro-inflammatory functions while not participating in fibrosis. According to the expression profile of cytokines and chemokines, M2b macrophages regulate the breadth and depth of immune and inflammatory responses (*Mosser & Edwards, 2008*). Exosomes from M2b macrophages were found to significantly reduce the severity of dextran sulfate sodium (DDS)-induced colitis in mice and inhibit the expression of key cytokines associated with colitis (IL-1, IL-6, and IL-17A) (*Yang et al., 2019*). Futhermore, the study of *Frodermann et al. (2016)* suggest that M2b macrophages may have an anti-atherosclerosis effect by inhibiting inflammatory Ly-6Chi monocytes, Th1 and Th17 cells infiltration. However, to our knowledge, experimental studies of the effect of M2b macrophages on PASMCs have not been performed. Our results showed that after treatment with M2b supernatant, the proliferation, migration, and apoptosis resistance of PASMCs were inhibited. Interestingly, previous studies have also suggested a role of M2b macrophages in improving pulmonary remodeling in PAH. In experimental models of PAH, the transplantation of mesenchymal stem cells (MSCs) or treatment with MSC exosomes effectively reduced pulmonary artery pressure and reversed right ventricular remodeling, which is thought to be related to immune regulation through ameliorating injury and reestablishing homeostasis (*Baber et al., 2007*; *Kanki-Horimoto et al., 2006*). The effect of MSCs on the pulmonary macrophage phenotype is the basis of the therapeutic effect of MSCs in regulating pulmonary inflammation. Furthermore, MSCs and MSC-derived extracellular vesicles downregulated the production of IL-23 and IL-22

and induced the polarization of M2b macrophages (*Hyvarinen et al., 2018*; *Philipp et al., 2018*). This evidence suggests that M2b macrophages may play an essential role in reversing pulmonary vascular remodeling in PAH.

In the current study, to further investigate the mechanism of conditioned medium from M2b macrophages on PASMC apoptosis, we assessed the expression of Bcl-2 family proteins and cleaved caspase-9 using Western blot. The activation of caspase promotes the release of cytochrome c and apoptosis-inducing factors, which mediates the mitochondrial regulation of apoptosis (*Cui & Placzek, 2018*). Bcl-2 family proteins regulate this process. Proteins in the Bcl-2 family are divided into two major subfamilies based on their function: the anti-apoptotic proteins (Bcl-2, Bcl-xl, Bcl-w) and the pro-apoptotic proteins (Bax, Bak, Bid, Bim) (*Siddiqui, Ahad & Ahsan, 2015*). The inhibition of Bcl-2/Bcl-xL has been reported to promote PASMC death and reverse pulmonary vascular remodeling in rats (*Rybka, Suzuki & Shults, 2018*; *Suzuki et al., 2007*). Conditioned medium from M2b macrophages regulates the expression of the Bcl-2 family proteins in PASMCs to induce apoptosis.

It is known that the PI3K/Akt pathway can regulate the activity of Bcl-2 family members (*Liu et al., 2018b*). Furthermore, this pathway is an important signal transduction pathway in cells and is mainly involved in regulating cell growth, metabolism, differentiation, and apoptosis (*Yu & Cui, 2016*). Therefore, the activation of the PI3K/Akt signaling pathway provides PASMCs with an ideal environment to survive, leading to continued cell proliferation, migration, and resistance to apoptosis (*Fang et al., 2016*; *Li et al., 2016*). Our results showed that inhibition of the PI3K/Akt signaling pathway may inhibit the proliferation and migration of PASMCs and promote apoptosis in the M2b group. Furthermore, inhibition of the PI3K/Akt pathway may also regulate the expression of Bcl-2 family proteins, which suggests that conditioned medium from M2b macrophages regulates Bcl2 family proteins through the PI3K/Akt pathway. Previous studies have found that inhibition of the PI3K/Akt pathway suppresses the cell cycle progression and arrests PASMCs at the G0/G1 phase, which inhibits cell proliferation (*Zhang et al., 2012*). *Zhang et al. (2016)* found that inhibition of PI3K/Akt pathway by carvacrol can attenuate pulmonary vascular remodeling and promote PASMC apoptosis.

Because FoxO3a is a direct downstream target of Akt, we investigated the potential role of FoxO3a in PASMC survival. Interestingly, both conditioned medium from M2b macrophages and inhibitors of PI3K significantly inhibited the levels of phosphorylated FoxO3a. As a member of the FoxO transcription factor family, FoxO3a is associated with cell cycle regulation and apoptosis induction. It plays a vital role in cell proliferation, apoptosis, cell cycle arrest, cell senescence, cell differentiation, and DNA repair (*Liu et al., 2018a*). When the PI3K/Akt signaling pathway is inhibited, FoxO3a is dephosphorylated, remains in the nucleus, and binds to promoters of target genes such as Bim, Fas, and CC3 and induces apoptosis. However, if the PI3K/Akt signaling pathway is activated, the phosphorylation of FoxO3a is stimulated, and FoxO3a transfers from the nucleus to the cytoplasm, leaving its target genes, resulting in cell survival (*Liu et al., 2019*). In conclusion, conditioned medium from M2b macrophages may regulate the proliferation, migration, and apoptosis of PASMCs through the PI3K/Akt/FoxO3a pathway.

 

As our current understanding of the immunology of pulmonary artery hypertension improves, these advances may pave the way for the "next generation" of therapies. For this reason, immunoregulatory approaches have also been considered. Moreover, stem cell and progenitor cell-based therapies, such as MSCs, induced pluripotent stem cells (iPSs), endothelial progenitor cells (EPCs), and human amniotic epithelial cells (AECs), have shown promise in experimental models for the treatment of PAH (*Willis et al., 2018*). This cell-based approach is useful in providing protection, anti-inflammatory, regeneration, and improved function. Thus, understanding the role of M2b macrophages in pulmonary pathophysiological immune responses at the molecular level lays the foundation for immunoregulatory approaches and cell-based therapies.

There are some limitations to the current study. First, we did not set other macrophage subsets as controls to demonstrate the specificity of the beneficial effects of M2b macrophages. Second, we are still exploring which components of the supernatant secreted by M2b macrophages are functional to further investigate the mechanism by which it works. Third, this study lacks animal experiments to verify the *in vivo* effect of M2b macrophages in PAH.

## CONCLUSIONS

In summary, we found that conditioned medium from M2b macrophages can inhibit the proliferation and migration of PASMCs and reverse their resistance to apoptosis. Furthermore, regulation of apoptosis carried out by conditioned medium from M2b macrophages is accomplished through the control of Bcl-2 family proteins. Conditioned medium from M2b macrophages may play a role in PASMCs by inhibiting the activation of the PI3K/Akt/FoxO3a pathway and its downstream target molecules. Our study highlights the effect of M2b macrophages on PASMCs in the hope of exploring their potential therapeutic role in improving pulmonary vascular remodeling in PAH. Activating M2b macrophages by transplantation or other means may be a new way to improve pulmonary vascular remodeling. M2b macrophages are promising for the treatment of pulmonary artery hypertension.

## ACKNOWLEDGEMENTS

The authors thank Keke Wang, Jiaxing Huang, and Jiawen Li (Department of Cardiac Surgery, The First Affiliated Hospital of Sun Yat-Sen University, Guangzhou, China) for sample collection.

### Funding

This work was supported by the National Key R&D Program of China [NO. 2017YFC1105000] and the National Natural Science Foundation of China [NO. 81570039, 81770319]. The funders had no role in study design, data collection and analysis, decision to publish, or preparation of the manuscript.

## Grant Disclosures

The following grant information was disclosed by the authors:
National Key R&D Program of China: 2017YFC1105000.
National Natural Science Foundation of China: 81570039, 81770319.

## Competing Interests

The authors declare there are no competing interests.

## Author Contributions

- Suiqing Huang conceived and designed the experiments, performed the experiments, analyzed the data, prepared figures and/or tables, authored or reviewed drafts of the paper, and approved the final draft.
- Yuan Yue conceived and designed the experiments, performed the experiments, analyzed the data, prepared figures and/or tables, and approved the final draft.
- Kangni Feng, Xiaolin Huang and Huayang Li performed the experiments, prepared figures and/or tables, and approved the final draft.
- Jian Hou and Song Yang analyzed the data, authored or reviewed drafts of the paper, and approved the final draft.
- Shaojie Huang performed the experiments, prepared figures and/or tables, authored or reviewed drafts of the paper, and approved the final draft.
- Mengya Liang analyzed the data, prepared figures and/or tables, authored or reviewed drafts of the paper, and approved the final draft.
- Guangxian Chen and Zhongkai Wu conceived and designed the experiments, authored or reviewed drafts of the paper, and approved the final draft.

## Animal Ethics

The following information was supplied relating to ethical approvals (i.e., approving body and any reference numbers):

The Animal Ethical and Welfare Committee of Sun Yat-sen University approved this study (DB-17-0506).

## Data Availability

The raw measurements are available in the Supplemental File.

## Supplemental Information

Supplemental information for this article can be found online at http://dx.doi.org/10.7717/peerj.9110#supplemental-information.

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
