# Peer review of "Conditioned medium from M2b macrophages modulates the proliferation, migration, and apoptosis of pulmonary artery smooth muscle cells by deregulating the PI3K/Akt/FoxO3a pathway"

_PeerJ, doi:10.7717/peerj.9110_

## Round 0.1 · original submission · Minor Revisions

As you will see, although both reviewers have provided a list of things to attend to - none of these should be difficult. Both have asked for more detail on quantification and statistics which we think are important. I think you should also make an effort to clearly elaborate upon the novelty and importance of this study to provide a better contextualisation of the study. Thanks for submitting this PeerJ and I hope you find these reports useful.

·

Basic reporting

This article is clearly and unambiguously written and uses professional English language throughout. Sufficient reference to prior knowledge in the field is given. Article structure and production is professional, the raw data has been shared and the results are relevant to the hypotheses.

Experimental design

This article contains original primary research within the scope of PeerJ. The research question is well-defined, as is the knowledge gap. The methods need extra details adding in order to replicate, and rationale for some of the techniques needs to be provided (please see comments below). Ethics are fine.

Validity of the findings

The impact and novelty of this article are not clearly communicated - the mechanistic insight is an association rather than a definitive cause-effect relationship and so the discussion and conclusion needs to be presented as such. Underlying data has been provided and is sound, however some revision of statistics in Figure 1 needs to be completed (please see comments below).

Additional comments

This article is well written in an easy-to-understand manner and contains sufficient new experimental data. I have a number of comments that should be addressed prior to publication.

1. Discussion: The proposed pathway of M2b inhibiting PI3K / Akt / FoxO3A causing reduced proliferation, migration and increased apoptosis has not been definitively proven. Further experiments e.g. showing that restoration of these signalling pathways abrogates the functional effects of M2b would need to be completed.

2. Methods: How many SD rats were used for SMC isolation and how many for BMDMs? Were any used for both? What was the age of the SD rats at MCT injection?

3. Methods: Line 153 suggests a 1:1 ratio of the number of SMC and macrophages used in the experimental studies. Does this correlate with the ratio of SMC and macrophages in vivo in PAH? Also, does the n number refer to the number of SMC donors, macrophage donors or both?

4. Results: Line 324 says PI3K was 'blocked' however the data in Figure 5 shows only a ~30% inhibition of PI3K with the LY294002.

5. Statistics: The data in Figure 1 is a comparison of two groups and so cannot be analysed by one-way ANOVA. Please re-analyse using two-tailed t-test.

6. Methods: Fibroblasts can also stain positively for a-SMA. To definitively prove the presence of SMC, co-staining for both a-SMA and SM-MHC should be observed.

7. Methods: Was the cell cycle of SMC controlled by serum-starving prior to the proliferation experiments being conducted? If not, how do you know that changes in proliferative rate are not simply a result of different SMC populations being in different stages of the cell cycle? Would you expect these SMC to proliferate within 24h?

8. Methods: How can you be sure that the observed changes in migration at 24h are not due to the changes in proliferation? Also, the chemoattractant gradient in transwell chambers only lasts for ~8 hours so will not be present at 24h.

9. Figures: Figure 3A has an apoptosis rate of 0.05-0.18, whereas Figure 3B has an apoptosis rate of 5-15%. Please can you explain this discrepancy?

10. Figures: Figure 1A, Figure 2B and 2C, Figure 3A and Figure 5C all need scale bars, and Figure 1 needs n numbers.

11. Figures: The bands for Bcl-2 and Bcl-xl in Figure 3C are overexposed.

12. Figures: Bands 3 and 4 in Figure 5A for p-PI3K do not match with the densitometry.

·

Basic reporting

The paper is clearly written and generally easy to follow. Literature is supportive of the manuscript scope with good coverage and breadth of the field.

I have provided minor comments for improvements to introduction, results, figure presentation and processing which i feel with strengthen the manuscript.

Introduction:
The introduction easy to understand – suggestions for text for improvement are below:
Text Minor Points:
Line 63: consider re-wording the opening line – the Tuder et al., (Tuder et al., 1994)’ is repetitive. Consider placement of reference here at the end of statement.
Line 73: can the authors give examples of the drugs used to target macrophages here.
Line 81: ROS in full first before abbreviation
Line 88: M2b macrophages are ‘recently regulated macrophages’ – the wording needs to be more clearly articulated.
Line 101: ‘We also explored the possible mechanisms’ – I feel this should be more elaborate.
Line 387: define DDS

Results Text:
Section 254-267: This section needs to be more clear. The authors use the supernatants from M0 and M2b for their assays. For me, the effects observed in this section is therefore the result of the products released into the M0 and M2b supernatant, not the M0 and M2b directly themselves. I think this needs to be more carefully worded.

Line 266: Can the authors clarify if it is M2b macrophages themselves that inhibit proliferation / migration (as stated) or is it the products released into the supernatant from M2b macrophages (lines 257-258). This is linked to the lack of clarity in the section above. Can the authors comment here on this.

Results Figures:
• Can the authors provide molecular weights for all Western blot data throughout the paper to indicate the band sizes of the proteins being presented.
• The authors need to take care with the contrast adjustments and cropping made to blotting panels. Please refrain from cropping close to the bands. See comments below.

Figure 1: Please state the number of repeats carried out for these experiments in the legend.
Panel A: Please provide the quantification data to support the >80% αSMA expression statement. Scale bars are missing from the images. Please add this information to the image.
Can the author comment about the cells in the IF that did not stain for αSMC. Was any other cell marker used to confirm the other cell population?
Figure 2: Panel C. The background of transwell M0 image has been adjusted and is clearly different from control and M2b. Can the authors explain this?

Figure 3: Panel A. Apply scale bars to the images. The IF images are not clear what the arrows are pointing at. It would be better to apply the arrows to the merged image rather than DAPI with a zoom into a defined area. Whilst the authors provide an n number for this experiment, can the authors indicate in the legend the total number of PASMCs counted across the N=5.
Panel B: The flow cytometry figure needs to be bigger. The text size on the axis is small and cannot be easily viewed.
Panel C: the blotting signal in the representative blots for Bcl-2 and Bcl-xl looks saturated which makes it difficult for reliable densitometry data to be quantified. Please provide representative blots that are not saturated (less exposed).
Panel C: The cleaved caspase band has almost been cropped from the blot. Please adjust this blot panel to that the band is not at the edge of the panel and more centrally aligned.

Figure 6: Panel A: Flow cytometry quadrant plots are too small which makes the axis and information difficult to read.
Panel B: Bax blot panel is cropped too close to the top of the protein bands. Please adjust.

Experimental design

Overall the methods are clear with sufficient details to repeat experimentation. Additional information is required at key parts prior to publication. See below:

Minor points:
Line 105: Age of the mice should be included and explanation of why only make mice were used for this study.
Line 152-153: ‘The stimuli of a certain number of PASMCs…… by the same number of macrophages’. Clarity is needed here in this statement. This needs to be more scientific. The authors need to define the cell number here.
Line 239: ‘The cells were then tested’. This needs to be more clear – what assays? More details are needed here.

Validity of the findings

The study reveals interesting results that demonstrate differences in M0 to M2b macrophage upon PASMC function. The findings in the manuscript would be of interest to the field. The authors clearly acknowledge the limitations to the study with the secreted products of M2b macrophages being unknown. Knowledge of these products would add impact to the paper however I do recognise that this would mean a significant volume of additional work which I feel would be unfair to request the authors to do. That would be a whole new paper in itself. I do request that the authors be more clear in the conclusions/abstract. I feel the authors needs to state that it is the supernatants from the M2b (and therefore the unknown products that they secrete) that give rise to the effects. At the moment it reads as if it is the M2b macrophages directly.

With the minor changes that i have recommended, I feel that the paper would be of interest to the readership of PeerJ.

Additional comments

No further comments to include. Please see individual comments related to the different section. Thank you for submitting your work to PeerJ. It was a pleasure to peer review your paper and I hope the comments can help you improve areas of the manuscript.

---

## Round 0.2 · accepted · Accept

Thank you for taking the time to carefully address all the points raised at review. I look forward to seeing the article on-line shortly.